# Further support for aneuploidy tolerance in wild yeast and effects of dosage compensation on gene copy-number evolution

Audrey P Gasch[1,2]*, James Hose[1], Michael A Newton[2,3], Maria Sardi[1], Mun Yong[1], Zhishi Wang[3]

[1]Laboratory of Genetics, University of Wisconsin-Madison, Madison, United States; [2]Genome Center of Wisconsin, University of Wisconsin-Madison, Madison, United States; [3]Department of Statistics, University of Wisconsin-Madison, Madison, United States

**Abstract** In our prior work by Hose *et al.*, we performed a genome-sequencing survey and reported that aneuploidy was frequently observed in wild strains of *S. cerevisiae*. We also profiled transcriptome abundance in naturally aneuploid isolates compared to isogenic euploid controls and found that 10–30% of amplified genes, depending on the strain and affected chromosome, show lower-than-expected expression compared to gene copy number. In Hose *et al.*, we argued that this gene group is enriched for genes subject to one or more modes of dosage compensation, where mRNA abundance is decreased in response to higher dosage of that gene. A recent manuscript by Torres *et al.* refutes our prior work. Here, we provide a response to Torres *et al.*, along with additional analysis and controls to support our original conclusions. We maintain that aneuploidy is well tolerated in the wild strains of *S. cerevisiae* that we studied and that the group of genes enriched for those subject to dosage compensation show unique evolutionary signatures.

*For correspondence: agasch@wisc.edu

**Competing interests:** The authors declare that no competing interests exist.

## Introduction

In our previous work (*Hose et al., 2015*), we reported from a genome-sequencing survey that aneuploidy was frequently observed in wild strains of *S. cerevisiae*, and that 10–30% of amplified genes, depending on the strain and affected chromosome, show lower-than-expected expression in naturally aneuploid isolates compared to isogenic euploid controls. We argued that this group is enriched for genes whose expression is regulated by one or more modes of gene-dosage compensation, which we validated at several genes. Importantly, the group of genes we identified shows non-random associations, including enrichment for specific functional groups, enrichment for genes that are toxic when over-expressed in the lab strain, and a higher rate of gene copy-number variation (CNV) despite higher constraint on expression variation in wild isolates of yeast. A main point of our paper was that dosage compensation at select genes has an important evolutionary effect, by buffering expression against CNV in wild strains.

Torres *et al.* take issue with the conclusions in our manuscript, arguing that a) wild strains of yeast are not tolerant of aneuploidy and our results emerge as an artifact of culturing wild strains in the lab, and b) errors in our statistical analysis explain the reduced expression of amplified genes and instead there are no genes subject to dosage-responsive control. We disagree with these assertions and provide additional computational analysis that support our original claims.

**eLife digest** Cells package their DNA into structures called chromosomes. Sometimes when a cell divides, it fails to allocate the right number of chromosomes to each new cell and so they end up with too many or too few chromosomes. The extra copies of the genes on an additional chromosome can be harmful to the cells, because the levels of the proteins encoded by those genes may rise abnormally.

Some organisms counteract the harmful effect of having additional chromosomes through a process called dosage compensation. Proteins are produced using genetic information via two steps: first a gene's DNA sequence is copied into a molecule of RNA, which is then translated into a protein. Dosage compensation can inactivate single genes or whole chromosomes via various means to ensure that the levels of RNA expressed remain normal, even in the presence of extra genes.

In 2015, researchers from the University of Wisconsin-Madison reported that dosage compensation occurs in wild strains of budding yeast and effectively protects against the harmful effects of having extra chromosomes. However, these findings conflicted with earlier studies of laboratory strains of this yeast, and earlier in 2016, other researchers re-analysed the previous study's data and challenged its findings.

Now, Gasch et al. – who conducted the work reported in 2015 – provide additional controls and computational experiments that support their original analysis. The latest analysis confirmed that the genes identified in the first study are indeed commonly duplicated in wild yeast populations, yet the expression of these genes remains controlled. This is consistent with a model of dosage compensation, for at least some of duplicated genes.

Gasch et al. believe that part of the difference in interpretation of the data relates to perspective. The challenging researchers tested to see if there was a mechanism of dosage compensation that acted across entire chromosomes, which is known to occur in the case of sex chromosomes in mammals. Gasch et al. on the other hand took a different approach and looked to identify effects at the level of individual genes.

Together, the analyses show that, while there is no evidence for a widespread mechanism, the expression of a select set of genes in wild yeast is consistent with gene-specific dosage compensation. Future work will now undoubtedly test the mechanisms behind the gene-specific effects, and explore why wild yeast strains are more tolerant to extra chromosomes than laboratory strains.

## Results and discussion

### Aneuploidy is relatively common and well tolerated in natural isolates of *S. cerevisiae*

Torres *et al.* argue that the wild strains analyzed in our study are not tolerant of aneuploidy, since the chromosome-wide average of the relative Illumina read depth measured for each amplified gene is not precisely 1.0, 1.5, or 2.0-fold higher than the euploid control (see Torres *et al.* Methods). They take this to reflect heterogeneous populations in which cells in the culture have lost the aneuploidy. However, this is not valid for several reasons. Due to technical biases in Illumina sequencing, it is highly unlikely that the mean value of relative gene copy number across whole chromosomes is a precise integer. Indeed, the plots shown by Torres *et al.* indicate the expected spread in relative read depth across the amplified chromosomes – similar to the spread in read counts of the euploid chromosomes – with mean values very close to the relative DNA abundance we reported. Furthermore, some genes on the chromosomes are not amplified (particularly those near the telomeres [*Hose et al., 2015*]), which can also slightly reduce the mean value away from a precise integer.

Regardless of the precise mean copy number values, there can be no doubt from the figures presented by Torres *et al.* that for the strains we analyzed in our original paper, the vast majority of cells in each culture were aneuploid. We point out that several of the chromosomes and strains highlighted by Torres *et al.* (Figure 1D-F and Figure 2F-H) were not presented in our manuscript or used in any of our analyses (see Hose *et al.* Figure 1A for strains and chromosomes used in our

study). It is true that some chromosome amplifications, namely in sake strains, are variable (appearing or disappearing) across replicates, and thus these chromosomes (Torres *et al.* Figure 1F and 2F) were not considered as part of this work. The karyotype of these sake strains may indeed be somewhat 'unstable' at the culture level; however, the random appearance of extra chromosomes across replicates again suggests a low fitness cost to aneuploidy and an observable rate of mitotic errors (*Zhu et al., 2014*). Nonetheless, the vast majority of aneuploidies we reported are relatively stable and maintained at high frequencies over many generations and in the absence of any selection.

Instead, our data show that the wild strains we studied are tolerant of aneuploidy, and that it is the laboratory W303 strain that is highly aberrant. 1) By conservative estimation, 30% of the strains we sequenced are aneuploid – these strains were identified in an unbiased sequencing survey in which aneuploidy was not generated or selected for. 2) The aneuploid strains we studied show little growth reduction compared to isogenic euploid strains, both for naturally aneuploid isolates and strains for which we artificially generated aneuploidy (*Hose et al., 2015*). (In cases where we cited the specific growth rate, we always verified that aneuploidy remained at the end of the experiment by relative qPCR.) Thus, aneuploidy tolerance is not due to unusual adaptation in the lab. In contrast, the tetrasomic W303_Chr12-4n strain – carrying a chromosome reported to be one of the least toxic in this background (*Sheltzer et al., 2012*) – has a 70% reduction in growth rate compared to its isogenic control (Hose *et al.* Figure 5B). 3) While extra chromosomes can be lost stochastically, it generally took >200 generations of growth to detect significant chromosome loss in the wild-strain cultures we analyzed. W303_Chr12-4n cultures reproducibly lose the extra chromosome in one culture passaging (~20 generations). The extreme fitness defect incurred by the aneuploid W303 strain explains the rapid emergence of cells that lose the extra chromosome, since the euploid W303 grows nearly twice as fast and rapidly takes over the culture. 4) Aneuploid W303 cells show aberrant gene expression and strong activation of the environmental stress response (ESR, [*Gasch et al., 2000*]), regardless of the chromosome amplified (*Sheltzer et al., 2012*; *Torres et al., 2007*). But as we published, naturally aneuploid strains simply do not activate the ESR (Hose *et al.* Figure 2A), indicating that they are not experiencing significant stress compared to their euploid controls.

Taken together, these results strongly support our conclusion that aneuploidy is relatively well tolerated in wild isolates of *S. cerevisiae*, at least for the strains and chromosomes we studied, but not in W303 as previously published (*Sheltzer et al., 2012*; *Torres et al., 2007*; *2010*). It is certainly possible that some wild strains will not tolerate aneuploidy, and likely that some chromosome amplifications are more problematic than others, regardless of strain background. The prevalence of aneuploidy in our original study is consistent with other reports of aneuploidy in wild isolates (*Tan et al., 2013*; *Strope et al., 2015*; *Sirr et al., 2015*; *Muller and McCusker, 2009*) and certainly industrial strains (*Bakalinsky and Snow, 1990*; *Bond et al., 2004*; *Hadfield et al., 1995*). Aneuploidy commonly emerges in response to selective pressure (*Mulla et al., 2014*); interestingly, a recent study by Filteau *et al.* showed that aneuploidy was relatively frequent in a wild strain, but not a laboratory strain, subjected to the same selection conditions (*Filteau et al., 2015*). W303 harbors six auxotrophies, several of which affect but do not entirely explain the intolerance of Chr12 amplification (unpublished). Thus, while many nice studies have been done with the W303 laboratory strain, we caution against using it (or any single strain) as the sole representative of *S. cerevisiae* (*Gasch et al., 2016*).

## A substantial fraction of amplified genes show lower-than-expected expression

To examine expression effects in naturally aneuploid strains, we measured mRNA and DNA abundance in aneuploid isolates compared to closely related or isogenic euploid controls. We did three sets of analyses in Hose *et al.* We first surveyed expression in six naturally aneuploid strains compared to paired euploid relatives, in biological duplicate. Pooling data across the strains identified 838 out of 2,204 amplified genes (38% over all genes considered in the six strains) whose relative mRNA abundance was reproducibly lower than the relative DNA abundance measured in the strain pairs (Hose *et al.* Figure 4A). The reduced expression could be because of a response to the aneuploidy, a response to the increased gene copy number, or due to heritable polymorphisms that reduce expression – we emphasize the presentation in Hose *et al.* that these genes should not be taken as dosage compensated from this analysis alone.

To distinguish the above possibilities, we performed two analyses on isogenic strain pairs, in which the only difference between strains was chromosome copy number. First, we measured mRNA abundance (in biological duplicate) for three aneuploid strains and their isogenic euploids and compared relative mRNA abundance to relative DNA abundance measured in those strain pairs (outlined in more detail below). This identified 163 of 882 genes (or 18% of the total set of genes assessed across these three strains) with lower-than-expected expression in aneuploid cells (Hose *et al.* Figure 4B and Supplementary File 3). Second, we generated isogenic strain panels for two other naturally aneuploid strains, in which diploid cells carried two, three, or four copies of the amplified chromosome. Using a mixture-of-linear regressions (MLR) model, we defined genes whose relative mRNA abundance did not increase proportionately to relative DNA abundance measured in the aneuploid versus isogenic euploid strains. The MLR analysis identified 172 genes in this class out of 773 genes (or 22% of the genes assessed in these two strain panels, Hose *et al.*, Table 1, Class 3a). We then combined the gene groups from the two analyses of isogenic strain sets for downstream analysis. Because the paired strain analysis was less stringent (in part because only duplicates were analyzed), we required that genes be identified in both Figure 4B (isogenic strain pairs) and Figure 4A (non-isogenic strain pairs). This left 73 genes whose expression was lower-than-expected in *four biological measurements* from isogenic strain pairs plus the 172 genes identified by the more sensitive MLR analysis from the isogenic strain panels, for a combined total of 245 genes out of 1655 (15%) total genes assessed in the two analyses of isogenic strains. In Hose *et al.*, we cited that 10–30% of genes, depending on strain and chromosome, met our criteria. There was an error in the abstract of the published article that we request to be changed, where '>30%' should have read 'up to 30%' of genes may be subjected to dosage compensation. We regret the error, but point out that the correct values are cited clearly throughout the manuscript.

Torres *et al.* disagree with our analysis methods, stating that i) the data were misnormalized, ii) the thresholds used were not valid, iii) the mixture of linear regressions (MLR) model was inappropriate, and iv) we did not correct for false discovery. They further propose that expression differences in aneuploid strains fit a normal distribution across the transcriptome, which they take as a null model for no dosage compensation. Below, we briefly address each of these points in turn.

First, as published in our original manuscript, several normalization methods were compared to ensure accuracy, including RPKM, RPKM excluding the amplified chromosomes, and the most accurate method of normalizing based on the number of collected cells. The latter is done by doping a fixed number of *Schizosaccharomyces pombe* cells relative to carefully counted *S. cerevisiae* cells in the collection, such that the *S. cerevisiae* sequencing reads can be scaled according to the known distribution of *Sz. pombe* reads across the samples (see Hose *et al.* for specifics). This is the most appropriate method of normalization, since it makes no assumptions about the data; however, it is particularly challenging for wild yeast that are often flocculent and difficult to count. Nonetheless, for several of the strains we analyzed the data normalized by *Sz. pombe* 'spike-in' agreed as well with the same data normalized by RPKM as did biological replicates normalized by RPKM (*Table 1*). (The exception was YJM428_Chr16 for which the spike-in normalization was clearly off based on comparing calculated Chr16 relative abundance to qPCR-measured Chr16 relative abundance from the same culture pairs, not shown). Data values were otherwise similar for different normalization methods, indicating that RPKM is a valid approach (*Hose et al., 2015*). Importantly, for our final analysis the DNA and RNA samples were normalized with the identical RPKM procedure – this produced global data centers (*i.e.* mean $\log_2$ values across all measurements) that were very similar for both mRNA and DNA measurements, indicating that the data were normalized in a comparable manner. Thus, there is no evidence that misnormalization of the data dramatically skewed our results.

Second, the description of our analysis methods presented by Torres *et al.* does not recapitulate what was done in our manuscript. For isogenic strain pairs shown in Hose *et al.* Figure 4B, we generated biological duplicate RNA-seq replicates and single DNA-seq samples, where DNA and one RNA sample were taken from the same culture, for both the aneuploid and euploid strains. For each replicate, we measured the relative mRNA abundance for each gene in the aneuploid versus euploid strain, as well as the relative DNA abundance for that gene in the aneuploid versus euploid strain. We then defined a gene-specific threshold to identify genes with lower-than-expected mRNA abundance: we took the relative DNA abundance measured for a given gene, minus one SD of the chromosome-wide mean of relative DNA abundances (*Table 1*). We then identified genes whose relative

**Table 1.** The center (mean) of $\log_2$ distributions for mRNA or DNA ratios measured across all unamplified genes in isogenic aneuploid-euploid strain comparisons are shown. mRNA data from each strain was normalized by either 'spike-in' of *Sz. pombe* cells to the *S. cerevisiae* cell collections or by reads-per-kb per million mapped reads (RPKM) of *S. cerevisiae* genes only. Normalized data were then compared across strain pairs to provide a $\log_2$ ratio of relative mRNA or DNA abundance for each gene.

| | Mean log2 mRNA ratios (spike-in) | Mean log2 mRNA ratios (RPKM) | Mean log2 DNA ratios (RPKM) | 1 SD used for threshold |
|---|---|---|---|---|
| T73_Chr8-4n vs -2n rep1 | 0.08 | −0.019 | −0.072 | 0.197 |
| T73_Chr8-4n vs -2n rep2 | 0.102 | −0.020 | n.a. | |
| YJM428_Chr16-4n vs -2n rep1 | −0.401 | −0.037 | −0.121 | 0.168 |
| YJM428_Chr16-4n vs -2n rep2 | −0.457 | 0.004 | n.a. | |
| YPS163_Chr8-2n vs -1n rep1 | n.a.* | -0.146 | 0.045 | 0.242 |
| YPS163_Chr8-2n vs -1n rep2 | n.a.* | −0.135 | n.a. | |
| NCYC110_Chr8-3n vs -2n rep5 | 0.014 | −0.005 | n.a. | |
| NCYC110_Chr8-4n vs -2n rep5 | 0.273 | 0.186 | n.a. | |
| NCYC110_Chr8-4n vs -2n rep1 | n.a. | 0.058 | −0.076 | |
| NCYC110_Chr8-4n vs -2n rep2 | n.a. | 0.011 | −0.108 | |
| NCYC110_Chr8-4n vs -2n rep3 | n.a. | 0.078 | n.a. | |
| YPS1009_Chr12-4n vs -2n rep1 | n.a. | 0.014 | 0.003 | |
| YPS1009_Chr12-4n vs -2n rep2 | n.a. | -0.094 | −0.229 | |
| YPS1009_Chr12-4n vs -2n rep3 | n.a. | −0.184 | n.a. | |

*We attempted spike-in normalization for haploid YPS163-disomic ('2n') and -monosomic ('1n') strains but were unable to accurately count cells due to differences in flocculation across aneuploid-euploid strains. As described in **Hose et al. (2015)**, RPKM normalization produced data that agreed as well or better across replicates compared to spike-in normalization and in the case of YJM428_Chr16-4n agreed better with qPCR measurements of Chr16 abundance in the culture (not shown). Note data from NCYC110 replicate (rep) 5 were not used in the analysis but were generated during the Hose *et al.* manuscript revision stage for normalization controls. The SD of DNA ratios on the affected chromosome that were subtracted from gene-level measurements of relative DNA abundance (i.e. to generate the gene-specific thresholds, see text) are shown for reference where relevant.

mRNA abundance in the aneuploid versus euploid strain was lower than the gene-specific cutoff *in both biological mRNA replicates*. Comparing measured mRNA ratios to measured DNA ratios (as opposed to theoretical DNA ratios) is critical to capture systematic and stochastic variation in the measurements. Our thresholding method allowed us to account for gene-specific biases in sequencing counts while incorporating measurement noise (and minimizing sequencing costs). Importantly, to meet our criteria for downstream analysis, a gene must have also met these criteria in the comparable analysis of non-isogenic strain pairs (Hose *et al.* Figure 4A). Thus, each gene had to be expressed below the threshold in *four biological replicates*. We note that it would be inappropriate to analyze the mean gene-level values from small numbers of replicates, which was not done in our analyses. In attempt to estimate the false discovery rate (FDR), we performed random permutations on the data (see Methods). We estimate the FDR for this analysis to be below 15%.

Torres *et al.* argue that our approach for selecting thresholds is not valid, citing the high standard deviation (SD) for mRNA abundance levels (RPKM) across the transcriptome (Torres *et al.* Figure 8B). However, we did not use abundance values, but rather *relative* abundance across isogenic strains, and thus the SD of RPKM values across all transcripts is not relevant to our analysis. What is relevant

is the SD of *replicate* mRNA abundance ratios compared to the SD of the relative DNA values used to define the threshold applied to replicates. The average SD of the replicate mRNA ratios for each gene ranged from 0.12–0.3, for amplified genes and for unamplified genes. This was the same range as the SDs of the relative DNA abundance ratios (0.17–0.24) used to define the gene-specific thresholds (*Table 1*), which were applied to four biological measurements of relative mRNA abundance. Torres *et al.* attempt to estimate the false discovery rate of our method on permuted data; while their methods are not entirely clear, they identify hundreds more genes on randomized data than we did on real data, and the SD values cited in their Table 2 are not close to our values, suggesting critical differences in methods.

In a second analysis, we generated two sets of isogenic strain panels in which diploid cells carried two, three, or four copies of a chromosome. Triplicate mRNA and duplicate DNA measurements were generated for each strain in each panel, and the data were fit using a sensitive mixture of linear regressions (MLR) model, which did not use the same cutoff approach to call genes. In fitting the MLR model, we did not average the replicates to reduce data to three points in a scatterplot per gene – such an approach could be adversely affected by outliers. Instead, we used all replicate measurements on all genes simultaneously, contrary to the assertion in Torres *et al.* The model takes into account variation in the replicate mRNA measurements (see Hose *et al.* for details). For simplicity, we classified genes based on their maximum posterior probability, and no cutoff on the magnitude of the expression effect was used – indeed, we showed in the paper that many genes have only partially reduced expression (Hose *et al.* Figure 4 and 6). Though it was not a focus of the original report, the MLR mixture model allows for a conditional FDR assessment, following the approach in *Newton et al. (2004)*. The 142 genes called Class 3A in the YPS1009 panel correspond to 17% FDR; the 30 genes called Class 3A in the NCYC110 panel correspond to 5% FDR by this method.

An independent check on MLR computations comes by using ordinary linear regression on the same input data, and using the regression to test the null hypothesis (per gene) of no deviation from proportional expression (i.e. the null is intercept=0 and slope=1). Under normal theory, the likelihood ratio test statistic has a chi-squared distribution on 2 degrees of freedom. An estimate of the proportion of null genes comes by processing the collection of chi-square p-values through Storey's q-value calculator (*Storey, 2003*). For both strain panels, these null proportion estimates are remarkably close to the Class 1 proportion estimates from MLR (in the YPS1009 panel: 14% by q-value and 15% by MLR; in the NCYC110 panel: 7% by q-value and 8% by MLR, both among the set of genes showing linear relationships on the log scale mRNA versus DNA). The likelihood ratio approach does not offer a direct classification of non-null genes, in contrast to the favored MLR approach used in Hose *et al.*, but does provide a check on inferences about the breath of putative-dosage compensation effects. In summary, the two methods we used to identify genes expressed lower-than-expected in isogenic aneuploid versus euploid strains identified 245 genes for downstream analysis. This amounts to 10 to 30% of genes depending on the strain and affected chromosome, or 15% of all genes assessed, at an FDR between 5 and 17%.

Torres *et al.* perform their own analysis to identify potentially dosage compensated genes, focusing on global averages, correlations, and distributions across the chromosomes. They argue that the distribution of expression effects across each aneuploid strain versus its isogenic euploid fits a normal distribution, in which there is a similar number of genes with higher expression on the affected chromosome as genes with lower expression. The authors cite that this is the null expectation in the absence of *any genes* subject to dosage compensation. While such a global approach would capture chromosome-wide dosage compensation as seen in sex chromosomes, we argue that this approach will miss many individual genes potentially subject to dosage control.

As published in Hose *et al.*, there are genes on and off the amplified chromosome that have higher expression in the aneuploid strains. The MLR analysis reported nearly the same number of amplified genes with higher expression as with reduced expression (Hose *et al.* Table 1). We regret the magenta coloring to highlight amplified genes in Figure 4, which was added during the revisions to make a separate point and thus the genes were not selected with the same criteria as genes with lower expression. Nonetheless, as we describe in Hose *et al.*, the two tails of the distribution are enriched for different functional groups: whereas amplified genes with lower-than-expected expression are enriched for particular functional groups (see below) and for genes that are toxic when over-expressed in the lab strain, the set of amplified genes with higher-than-expected expression is enriched for genes encoding membrane and cell-surface proteins. As we cited, the higher-than-

expected expression of amplified genes (and of genes on unamplified chromosomes) is likely an indirect response to the known influence that ploidy has on cell size/shape and flocculence (*Wu et al., 2010*). Therefore, we do not believe that the distribution across all genes can be used to select individual genes subject to dosage compensation, nor did we focus on global averages or correlations. Instead, we identified individual genes that have the lower-than-expected expression phenotype and analyzed the pooled set to explore enriched features, as described below.

## Unique features of selected genes suggest the group is enriched for dosage-compensated genes

A major point of our original manuscript that was not addressed by Torres *et al.* is that the genes we identified through the above analyses are strikingly enriched for unique patterns. The set of genes with lower-than-expected expression in aneuploid strains is strongly enriched for certain functional groups, including genes known to feedback on their own expression (see more below), genes that are toxic when over-expressed, and genes that are under higher expression constraint but display elevated copy number variation (CNV) in wild populations (*Hose et al., 2015*). Such non-random associations are unlikely to occur if the gene selection was truly random and driven by noisy data. We ruled out the possibility that these genes reflect a common, indirect response to aneuploidy. However, it is true that a subset of the genes we identified could be responding transcriptionally to the particular chromosome amplified, rather than the gene's copy number *per se*. We attempted to minimize these effects in our original downstream analysis by pooling amplified genes from five aneuploid strains and across three amplified chromosomes. Here, we provide an additional series of controls to show that the trends seen for the gene group we identified cannot be explained as indirect transcriptional responses to aneuploidy.

As presented by Torres *et al.*, chromosome-specific responses to aneuploidy should affect groups of functionally related genes, whether or not the genes are amplified. In an attempt to remove amplified genes that may be a part of such indirect responses, we first identified *unamplified* genes in each aneuploid strain with significantly reduced expression compared to its isogenic euploid (FDR 0.01) and identified enriched functional groups (see Methods). We then removed from consideration all amplified genes in that strain belonging to any of those strain-specific functional groups. (Note: we consider cytosolic and mitochondrial translation factors as separate groups). This analysis removed ~30% of the genes classified as potentially dosage compensated in the original analysis; the majority of these were linked to mitochondrial function and is consistent with the common mitochondrial response we reported (even though, interestingly, the removed genes were not affected in multiple aneuploid strains). We note that our procedure may remove genes that are legitimately dosage compensated, especially those that show only partial compensation, which could cause indirect effects on unamplified genes in the same functional category (*e.g.* partial compensation of a transcriptional repressor).

The revised set of amplified genes with lower-than-expected expression remains enriched for functional groups that cannot be explained by common or chromosome-specific responses, including cytosolic RPs (p=6e-7), other proteins involved in ribosome biogenesis (p=5e-4), and an overlapping gene set linked to translation (p=3e-6). RPs are known to feedback on their own expression (*Warner et al., 1985*; *Vilardell and Warner, 1997*; *Dabeva and Warner, 1987*; *Dean et al., 1981*; *Tsay et al., 1988*), and we validated the trend at two genes (*Hose et al., 2015*). The revised group is also weakly enriched for sequence-specific DNA binding proteins (p=0.006), including several transcription factors that bind their own promoters and/or regulate their own expression (including, Hap1, Stb5, Bdf1, and Rsc30 [*Venters et al., 2011*; *Deckert et al., 1995*; *Hon et al., 2005*; *Denby et al., 2012*]). Importantly, these enrichments remained significant when genes from each chromosome were held out (p<0.007, except for one case where DNA binding protein enrichment p=0.04), showing that the result is not skewed by one particular aneuploidy. As cited in Hose *et al.*, these categories are not enriched among unamplified genes whose expression is affected by the common aneuploidy response (*Hose et al., 2015*). Thus, our analysis is clearly enriching for genes subject to dosage compensation.

All of the evolutionary signatures we reported in the original manuscript remain statistically significant on this reduced gene group (*Figure 1*). The revised gene set enriched for dosage-compensated genes displays significantly higher levels of CNV in *S. cerevisiae* populations, compared to all genes and compared to amplified genes with proportionately elevated expression (*Figure 1A*). They

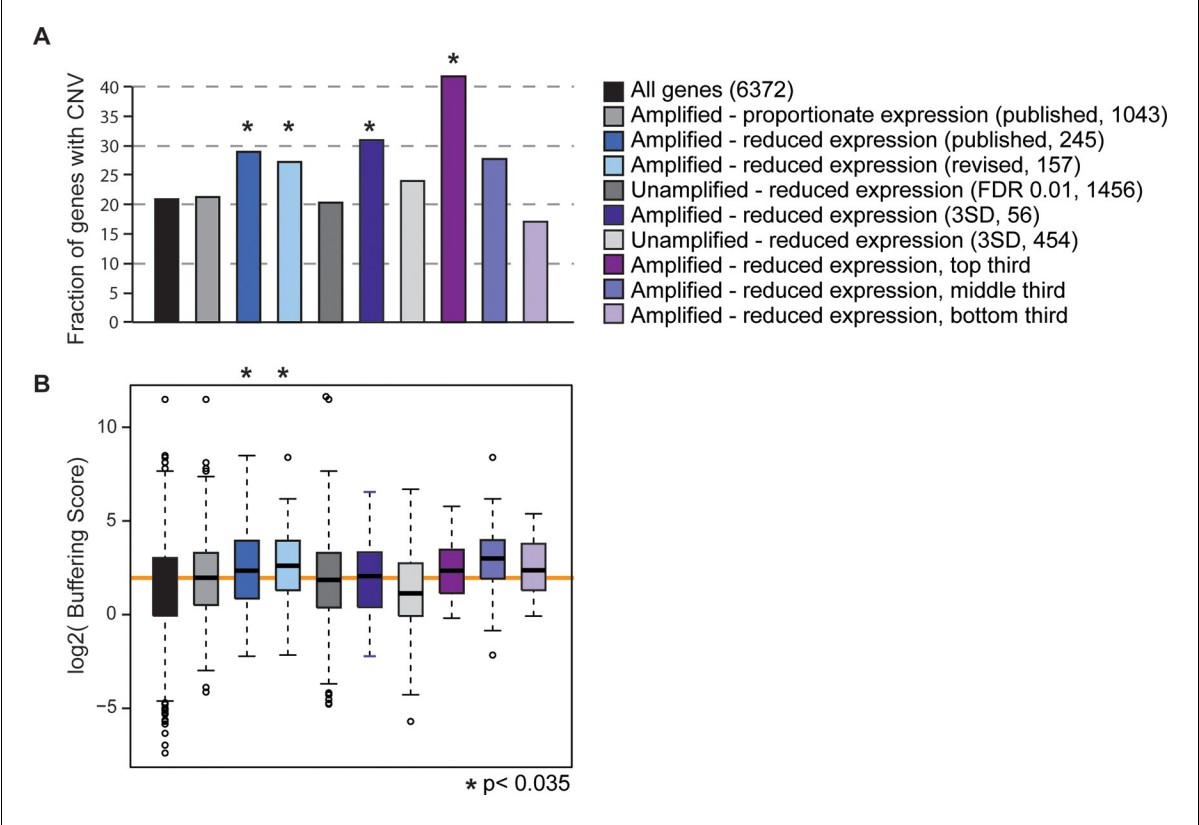

**Figure 1.** Gene sets enriched for dosage-compensated genes show unique signatures. Gene sets and the numbers contained within them are indicated by the key. The revised list of amplifed genes with lower-than-expected expression were partitioned into thirds, based on genes with the greatest (top third) or smallest (bottom third) reduction in expresion compared to expectation. (**A**) The fraction of genes in each group for which at least three of 103 strains showed gene amplification. (**B**) The distribution of Buffering Scores for genes with CNV. Here, Buffering Score represents the number of 103 strains with a gene duplication divided by expression constraint ($V_g/V_m$, see Hose *et al.* for details). Higher values indicate a higher propensity for CNV despite expression constraint. The orange line indicates the median value for amplified genes with proportionate expression as a reference point. Asterisks indicate statistical significance (p<0.035) compared to the amplified genes with proportionate expression. In some cases, the trends were consistent but not significant (likely owing to small sample sets).

also have significantly higher CNV buffering scores (*Figure 1B*, see Methods and (*Hose et al., 2015*)), indicating a higher level of CNV despite increased expression constraint. If these trends are driven by genes subject to repression or indirect effects of aneuploidy, we would expect to see the same patterns for *unamplified* genes that show reduced expression compared to euploid controls. But this is not the case: unamplified genes with reduced expression (identified either by *edgeR* or by the identical methods used to call dosage-compensated genes) show no increase in CNV propensity or buffering scores compared to the control group. The trends for the affected amplified genes remain if we define genes using a more stringent threshold to call affected genes, if we remove from the analysis all genes belonging to any of the enriched functional groups, or if we separately hold out each amplified chromosome from the analysis (see Methods). A final prediction is that genes with larger effect sizes in terms of the level of compensation should have more striking evolutionary patterns, and this is indeed the case (*Figure 1*). Therefore, the results we report cannot be explained as a secondary transcriptional response to the amplified chromosome – we believe that the group is legitimately enriched for genes subject to dosage control and that dosage compensation has an important effect on genome evolution.

Our original intention in Hose *et al.* was not to claim that dosage compensation is 'widespread' in *S. cerevisiae*, but rather that it exists at a significant number of genes and has an important role in evolution. We certainly agree with Torres *et al.* that dosage compensation does not function at most amplified genes, as seen for sex chromosomes in other organisms. Our goal was not to define

individual genes subject to dosage compensation, but rather to investigate the group – we caution that no single gene from our lists should be taken as dosage compensated without orthogonal evidence. In the end, the revised gene set we identified here at the highest confidence encompasses 157 (13%) genes out of 1,243 interrogated across all the isogenic strain sets. It is true that some of the genes we identified, particularly those with only subtly reduced expression patterns, are false positive calls from our analysis. However, the number of genes we identified is inline with the 14% of dosage compensated genes reported by Springer and colleagues (*Springer et al., 2010*) (using GFP reporters for 730 genes interrogated) and is consistent with (but on the lower end) our original report of 10–30%. Future work will undoubtedly clarify the precise number and identity of genes subject to dosage compensation, as well as the genetic basis for phenotypic differences in aneuploidy tolerance in laboratory versus wild strains.

## Methods

In the process of this work, we verified our original analyses to define gene groups. We estimated the FDR of our thresholding method as follows: 1) We calculated the $\log_2$ ratio of mRNA in aneuploid versus euploid cells, minus the corresponding $\log_2$ ratio of relative DNA abundance for that gene minus 1SD of the chromosome-wide mean of relative DNA values. A negative value indicates that the relative mRNA expression was below the relative DNA copy number minus 1SD. 2) These difference values were randomly permuted with regard to the gene labels, within each of two replicates in both the isogenic and non-isogenic strain pairs. 3) We calculated the number of genes for which all four of the randomized data values were negative, over 10,000 iterations. The FDR (average number of genes identified from 10,000 iterations on random data divided by the number of genes identified in real data) was calculated to be below 15.0%. We note that some fraction of true positives will meet the threshold in randomized trials.

We also generated a revised list of putatively dosage compensated genes as follows: Unamplified genes with significant expression differences in each strain were identified using edgeR (*Robinson et al., 2010*) comparing RPKM values in aneuploid versus isogenic diploid strains. Biological duplicate RNA-seq data were analyzed for T73_Chr8-4n versus T73_Chr8-2n, YJM428_Chr16-4n versus YJM428_Chr16-2n, and YPS163_Chr8-disomic versus YPS163_Chr8-monosomic; biological triplicates were analyzed for YPS1009_Chr12-4n versus YPS1009_Chr12-2n and for NCYC110_Chr8-4n versus NCYC110_Chr8-2n (see Hose *et al.* for all other details). Unamplified genes with an FDR <0.01 (*Robinson et al., 2010*) and a negative mean $\log_2$(fold-change) in aneuploid versus euploid expression were taken as 'lower expressed'. Functional enrichment was done on the set of lower-expressed unamplified genes for each strain separately, using the program FunSpec (*Robinson et al., 2002*) and taking p<0.0004 as significant. Amplified genes from that strain belonging to any of the strain-specific enriched functional categories (excluding those based on cellular localization) were removed from the original list of putatively dosage compensated genes. Note that we considered cytosolic and mitochondrial RPs and translation factors as separate groups. One of the strains (YJM428_Chr16-4n) had a substantial secondary response at unamplified genes, amounting to two thirds of the 1,456 lower-than-expressed unamplified genes pooled across the isogenic strain pairs – the effected genes from YJM428_Chr16-4n were admittedly enriched for translation factors and rRNA biogenesis genes, causing the removal of amplified Chr16 genes in those categories from the list of affected genes. Nonetheless, the functional categories described in the text remained significant for the remaining 157 genes with reduced expression in aneuploid strains. For analysis in *Figure 1*, the genes were ranked and partitioned into thirds based on the magnitude of expression reduction ('effect size', where 'top third' represents amplified genes with the strongest reduction in expression), taken as the average relative mRNA ratio for the isogenic aneuploid and euploid strains compared to the relative DNA ratio between those strains.

As a separate approach to call putatively dosage compensated genes, we used the same methods described for Figure 4B of Hose *et al.*, except that we selected genes whose relative mRNA values were less than the relative DNA ratio measured for that gene minus 3 SD of the chromosome-wide mean in that strain. A gene had to be below that gene-specific threshold in all (two or three where relevant) biological replicates of isogenic strain pairs. This identified 56 genes across the five strains. We applied an identical method to identify non-amplified genes with reduced expression; for this, we used the same SD as applied for amplified genes (although this was generally very similar

to the SD for all genes on unamplified chromosomes). This identified a total of 454 unamplified genes with lower expression ('3SD' in *Figure 1A*).

We plotted the fraction of genes for which at least three of 103 strains previously analyzed had evidence of gene amplification (see Hose *et al.* for details). Trends were similar when plotting the fraction of genes in which at least two strains displayed CNV. For simplicity, the buffering score plotted in *Figure 1* represents the number of 103 strains with CNV divided by the expression constraint score $V_g/V_m$ (see Hose *et al.* for details). We performed a variety of other controls, ensuring that trends were the same when each chromosome was held out separately, when genes belonging to any functional group were held out, and using different cutoffs for CNV. The trends were consistent in all cases (although not always statistically significant owing to small datasets in some cases).

## Acknowledgements

We thank members of the Gasch Lab for critical reading. This work was supported by grants from the National Institutes of Health (R01GM083989 to APG and U54AI117924 to MAN).

## Additional information

### Funding

| Funder | Grant reference number | Author |
|--------|------------------------|--------|
| National Institutes of Health | R01GM083989 | Audrey P Gasch |
| National Institutes of Health | U54AI117924 | Michael A Newton |

The funders had no role in study design, data collection and interpretation, or the decision to submit the work for publication.

### Author contributions

APG, MAN, Conception and design, Analysis and interpretation of data, Drafting or revising the article; JH, MS, MY, ZW, Analysis and interpretation of data, Drafting or revising the article

### Author ORCIDs

Audrey P Gasch, http://orcid.org/0000-0002-8182-257X

## Additional files

### Major datasets

The following previously published datasets were used:

| Author(s) | Year | Dataset title | Dataset URL | Database, license, and accessibility information |
|-----------|------|---------------|-------------|--------------------------------------------------|
| Hose J, Yong CM, Sardi M, Wang Z, Newton MA, Gasch AP | 2015 | RNA-seq data from aneuploid yeast strains | http://www.ncbi.nlm.nih.gov/geo/query/acc.cgi?acc=GSE61532 | Publicly available at the Gene Expression Omnibus (accession no. GSE61532) |
| Hose J, Yong CM, Sardi M, Wang Z, Newton MA, Gasch AP | 2015 | Genomic DNA-seq data from aneuploid yeast strains | http://trace.ddbj.nig.ac.jp/DRASearch/study?acc=SRP047341 | Publicly available at DNA Data Bank of Japan (accession no. SRP047341) |

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
