## [Decision Letter]

Thank you for submitting your work entitled "Further support for aneuploidy tolerance in wild yeast and effects of dosage compensation on gene copy-number evolution" for consideration by *eLife*. Your article has been reviewed by three peer reviewers, and the evaluation has been overseen by Duncan Odom as the Reviewing Editor and Randy Schekman as the Senior Editor.

The reviewers have discussed the reviews with one another and Dr Odom, the Reviewing Editor, has drafted this decision to help you prepare a revised submission. Despite the criticisms below, all reviewers in this *eLife* discussion board unanimously shared a commitment to allowing you a fair and open response in print to the scientific challenge offered by Torres et al.

Because your submission is an author-response to Torres et al., a manuscript that is currently in press at *eLife* challenging your prior work, I have appended the complete reviews from three independent reviewers, as well as one notable (anonymized) comment from our discussion. Note that lightly edited versions of these reviews will be available for the community to see. There were a number of criticisms of your lines of reasoning and the clarity of your presentation, as well as suggested experiments by Reviewer 1.

For re-submission of this Short Report, the only required changes are:

a) To revise the clarity and argument structure, both of which were considered at times weak by all the reviewers

b) Slowly cross-check all your numbers, as well as your references to numbers in your original work, and explain any discrepancies clearly.

c) Rewrite the Abstract to clarify, per Reviewer 3.

d) Revise your Discussion to admit possible weaknesses in your analyses. Do please consider including some or all of the following points: that the number of replicates used may be too low to reliably use means, that some of the strains may well be unstable, and all things considered, the number of genes that are dosage-compensated could be quite small – and are likely to be smaller than claimed in the original Hose paper.

Dr Odom, the Reviewing Editor, suggests that careful (and extensive) experimentation that begins from the types of experiments suggested by Reviewer 1 could serve as the nucleus of a more comprehensive story for a later submission that is entirely independent of the current dosage compensation discussion.

Reviewer #1:

“Further support for aneuploidy tolerance in wild yeast and effects of dosage compensation on gene copy-number evolution” is a rebuttal of a previous manuscript of Torres, Springer and Amon, which was itself a rebuttal of a paper published by Gasch, Hose et al. in 2015. It is a sad reality that the two groups were not able to find a common ground for a professional discussion that would result in a collaborative manuscript addressing the key issues. The reviewers and editors have to help to ensure that the rebuttal remains factual and professional, supported by clear arguments and genuine attempts to find a solution.

The rebuttal focuses on two main points. First, they argue that, oppositely to the claim of Amon and Torres, aneuploidy is indeed more tolerated in wild budding yeast strains. This claim is irrelevant, as Amon and Torres did not question the claim that wild yeast strains are more tolerant to aneuploidy (they questioned that this tolerance is due to the ability to compensate the gene dosage). In the second part, they argue that their analysis of dosage compensation of mRNA levels of amplified genes is correct and that a significant proportion of genes is compensated in wild yeast strains. This part is confusing and flawed. Several points need to be addressed:

1) Subheading “A substantial fraction of amplified genes show lower-than-expected expression”: It is questionable that the normalization strategies used for mRNA and genomics NGS data should be identical. RPKM is used exclusively for mRNA normalization, as far as we know. It is not clear how "careful cell counting and doping with *Schicosaccharomyces pombe* cells at the time of collection" should have helped to improve the normalization. This requires further explanation. Finally, log_2_ difference of paired RNA-DNA sample at -0.04 feels dangerously low – this similarity can rarely be found even between technical replicates. Using this precise number as a comparison between two very different sequencing experiments seems to be in stark discrepancy to the argument in paragraph one, subheading “Aneuploidy is relatively common and well tolerated in natural isolates of *S. cerevisiae*”, where the authors state that due to technical biases in Illumina sequencing the mean value isn´t a precise measurement and can´t be used to conclude on a unbalanced karyotype.

2) Paragraph four, subheading “A substantial fraction of amplified genes show lower-than-expected expression”: Definition of the cut off is the critical point. Why the cut off for mRNA levels should be based on DNA abundance is beyond our understanding – the variance of DNA abundance and variance of RNA abundance are usually very different. The rationale is not explained.

3) The next part of the paragraph brings a flurry of numbers that is difficult to follow. They first state that they identified 163 of 882 genes tested to be dosage compensated – and not 838, as stated by Torres. But in the Hose et al. manuscript, there is indeed the number 838, and the number 163 or 882 is nowhere to be found. They also claim that their subsequent analysis defined stringent criteria for dosage compensation to be met by 88 genes out of 738. They cite Figure 4A in Hose et al., but again, this number is not there. This means dosage compensation in 12% of amplified genes, but at 15% FDR. None of these numbers was mentioned in the previous manuscript.

4) Paragraph five, subheading “A substantial fraction of amplified genes show lower-than-expected expression”: Why the SD across all transcripts is not relevant is beyond our understanding. Both SDs across replicates and SD across all transcripts within one sample are relevant, each of them provides different information.

5) Paragraph six, subheading “A substantial fraction of amplified genes show lower-than-expected expression”: The criticism of MLR by Torres and Amon was based on the fact that only 3 samples were used for the analysis, which is not sufficient; here it is claimed that 12 samples were used – however, this number includes the biological replicates of 3 samples only, which is indeed not sufficient. All the additional analysis does not help the fact that the model is based on 3 samples.

6) They agree with Torres and Amon that the distribution of the gene expression of amplified genes is normal (not skewed), yet argue that the downregulated genes represent dosage compensation, whereas the upregulated genes are a part of the general response to aneuploidy – it is very unclear why each of these two outliers' groups should be interpreted differently. Their argumentation in the chapter "Unique features of selected genes...." is a prime example of scientific tautology: genes are dosage compensated because it is disadvantageous for the cells to overexpress them. Accordingly, for these genes a propensity to CNV is observed. They do not observe propensity to CNV for genes from these categories when they are present in normal copy numbers – but this is most likely because they are not overexpressed and therefore there is no need to select against them! They also never say what gene categories are downregulated among unamplified genes (likely the same categories as on the amplified chromosomes).

7) There are also genes that are up-regulated – the identified categories broadly agree with the previous gene categories identified to be upregulated in response to aneuploidy (they cite the work from Fink lab, which relates to tetraploidy, but see more relevant Amon, Torres, and Storchova lab results). Remarkably, the dosage compensated genes closely match the categories identified previously to be downregulated in response to aneuploidy – ribosome biogenesis, translation, transcription – in several different species (see Amon, Torres, Storchova, Foijer laboratories). This option should be considered.

8) In paragraph five, subheading “Unique features of selected genes suggest the group is enriched for dosage compensated genes”, they cite a paper by Springer et al. (Mol Sys Biol 2010) in support of the presented work. But Springer's manuscript is focusing entirely on dosage compensation on protein level, whereas Hose considers mRNA; of individual proteins only, measured one-by-one by FACS, whereas Hose et al. considers whole chromosomes only, and Springer analyzes loss of copies, whereas Hose analysis gain of gene copies.

In conclusion, the arguments do not seem strong enough to justify their analysis approach. Secondly, even if we would agree with the analysis, they found that the mRNA of 12 – 13% of genes might be dosage compensated at a FDR of almost 15% – a result that in fact says that there is no general dosage compensation.

Reviewer #2:

This short paper by Gasch, Hose and colleagues is a reply to a paper by Torres and colleagues, which is in turn a reply to a first paper by Hose, Gasch et al. in which they report that several wild yeast strains show high levels of aneuploidy and that the expression of some genes does not reflect their copy number changes, suggesting that these wild yeasts have some mechanisms of gene dosage compensation that reduces the expression effect of copy number variation (which may explain why such high levels of aneuploidy are tolerated).

The paper by Torres and coworkers notes two key issues in the work by Gash et al:

1) The copy number variation is based on changes in read depth of illumina genome sequencing. However, the read depth does not always change in integer numbers, which may indicate that only part of the population shows the CNV and that the populations are heterogeneous.

2) The statistical methods used by Hose, Gasch et al. are not correct – most notably, Torres and coworkers argue that Gasch et al. do not account for multiple testing and do not use appropriate error calculations, since the expression levels were calculated with very noisy mRNA abundance levels (RPKM).

Gasch, Hose et al. now argue that:

1) The non-integer changes in read depth are due to sequencing **bias**

2) The error calculations were OK since everything was based on (less noisy) relative abundance levels.

Major comments:

1) As I already indicated in my review of Amon et al., I am also puzzled by the non-integer changes in sequencing depth. I do think that this suggests that the strains could indeed be unstable. However, I also noted that we need to be sure that sequencing biases are not involved. I suggested a few relatively easy experiments to verify whether the populations are homogeneous and whether they show the expected aneuploidies. It seems easy enough to measure the copy number of a few key regions in a few key strains using qPCR, and to do this on a few different (small) colonies of the same culture. This would close the discussion.

2) I follow the arguments about the error calculations. Although I am not an expert in this area, I think that, Gasch, Hose and coworkers indeed used correct calculations, but their sample size (number of replicates) are very low. Again, a few extra experiments using qPCR would help to close the discussion.

3) The number of genes that may show dosage compensation (245) seems quite low, especially since many of these might be false (false discover rate…). Even if some GO enrichment is found in this set, it is not clear to me that the number is high enough to call this "widespread". My best guess is that strains become aneuploid and the relatively quickly acquire mutations that normalize the expression of a few key genes for which changes in expression are not tolerated. Whether this is special for wild yeasts and should be called "dosage compensation" is a difficult question (obviously, the dosage is compensated, but using the name also suggests a general, dedicated mechanism, and this may very well be lacking…)

Reviewer #3:

This manuscript by Gasch et al. is a rebuttal to the Torres et al. criticism of Hose et al. Both Hose et al. and Torres et al. were published in *eLife*. I will preface by saying that I firmly believe Gasch should have the opportunity to present a response (this manuscript).

At the heart of this dispute is to two points: namely (1) is aneuploidy relatively common in wild strains and well tolerated there? and (2) to what extent (fraction) do genes show a lower-than-expected expression?

Here Gasch provides a strong rebuttal of the Torres concerns regarding whether aneuploidy is common and tolerated in wild strains. In my opinion the Torres arguments on this point were weak and Gasch clearly articulates the relevant counter argument.

On the second point, namely to what extent do genes show a lower than expected expression, Gasch clearly outlines the details of the methods used in Hose and why the criticisms of Torres were off the mark at times. Gasch also correctly points out the Torres methods for estimating FDR were poorly described. Gasch provides additional details on the methods of Hose and additional, more stringent analysis that reaches many of the same conclusions of Hose.

One notable (anonymized and lightly edited) comment from the reviewers’ discussion:

On the second issue, the extent of dosage compensation, I think that there is a fundamental disconnect between the groups in how to assess dosage compensation. Gasch (in Hose and Gasch) seek to take a gene specific view that accounts for gene-to-gene variability. This is noble and philosophically sound, but fraught with problems they don't address – namely that they have way too small a sample size to *really* do this (as also noted by both other reviewers). The Amon approach is to take a more distribution-based approach – which is, in principle, better in low sample sizes *–* but as Gasch correctly points out (and was a major criticism of the Torres manuscript) *–* the Torres methods are so poorly described as to be impossible to assess. Even if you think the Gasch method has merit, in the end they do back peddle on the number of dosage compensated genes (in Gasch compared to Hose) leading one to suspect we are talking about something that is either a small class or even non-existent.

---

## [Author Response]

For re-submission of this Short Report, the only required changes are:

a) To revise the clarity and argument structure, both of which were considered at times weak by all the reviewers

b) Slowly cross-check all your numbers, as well as your references to numbers in your original work, and explain any discrepancies clearly.

c) Rewrite the Abstract to clarify, per Reviewer 3.

d) Revise your Discussion to admit possible weaknesses in your analyses. Do please consider including some or all of the following points: that the number of replicates used may be too low to reliably use means, that some of the strains may well be unstable, and all things considered, the number of genes that are dosage-compensated could be quite small – and are likely to be smaller than claimed in the original Hose paper.

We have revised the text in several places, providing additional detail as well as a new Table 1 to clarify our explanations and reasoning. We have also provided a more detailed summary of Hose et al. including the genes selected at each step and references to where those gene lists/data were reported in Hose et al. We double-checked the values from the original paper and clarified the Abstract as suggested by Reviewer 3. We hope that these changes have clarified our arguments in these places.

We have also revised the Discussion in several places as requested: We added a statement in paragraph two, subheading “Aneuploidy is relatively common and well tolerated in natural isolates of *S. cerevisiae*” that some of the strain karyotypes may be unstable; we added a sentence to the last paragraph that states that some of the genes on our list will be false positives, and another statement that our revised gene set is on the lower end of what we originally reported when considering all genes as a single group. These changes are in addition to a statement that individual genes from our list should not be taken as dosage compensated without orthogonal evidence. We also added a statement to paragraph five, subheading “A substantial fraction of amplified genes show lower-than-expected expression” stating that it is inappropriate to use mean gene-level expression values from small numbers of replicates, which was not done for any of our analysis.

Reviewer #1:

“Further support for aneuploidy tolerance in wild yeast and effects of dosage compensation on gene copy-number evolution” is a rebuttal of a previous manuscript of Torres, Springer and Amon, which was itself a rebuttal of a paper published by Gasch, Hose et al. in 2015. It is a sad reality that the two groups were not able to find a common ground for a professional discussion that would result in a collaborative manuscript addressing the key issues. The reviewers and editors have to help to ensure that the rebuttal remains factual and professional, supported by clear arguments and genuine attempts to find a solution.

We wholeheartedly agree with the reviewer on these points.

The rebuttal focuses on two main points. First, they argue that, oppositely to the claim of Amon and Torres, aneuploidy is indeed more tolerated in wild budding yeast strains. This claim is irrelevant, as Amon and Torres did not question the claim that wild yeast strains are more tolerant to aneuploidy (they questioned that this tolerance is due to the ability to compensate the gene dosage).

In fact, Torres et al. spend considerable effort to argue that the aneuploid strains are ‘unstable’ and therefore not tolerant to aneuploidy. As indicated by reviewer #3 below, our response to these points is strong and convincing.

1) Subheading “A substantial fraction of amplified genes show lower-than-expected expression”: It is questionable that the normalization strategies used for mRNA and genomics NGS data should be identical. RPKM is used exclusively for mRNA normalization, as far as we know. It is not clear how "careful cell counting and doping with Schicosaccharomyces pombe cells at the time of collection" should have helped to improve the normalization. This requires further explanation. Finally, log_2_ difference of paired RNA-DNA sample at -0.04 feels dangerously low – this similarity can rarely be found even between technical replicates. Using this precise number as a comparison between two very different sequencing experiments seems to be in stark discrepancy to the argument in paragraph one, subheading “Aneuploidy is relatively common and well tolerated in natural isolates of S. cerevisiae”, where the authors state that due to technical biases in Illumina sequencing the mean value isn´t a precise measurement and can´t be used to conclude on a unbalanced karyotype.

We apologize for the confusion, as the description of the doping normalization was included in the original Hose et al. manuscript. We have added a clearer description and referencing for this normalization procedure to paragraphs two and three, subheading “A substantial fraction of amplified genes show lower-than-expected expression”. We also added a table to present the centers (means) of log_2_ distributions for normalized data, which this reviewer may have misunderstood. The important points from this section are: 1) the most accurate method of *Sz. pombe* spike-in normalization gives comparable normalization to the more robust RPKM (aside of one strain, for which the spike-in normalization was clearly off, for reasons we describe in the text); 2) the center of the distributions for RPKM-normalized mRNA ratios was very similar to the center of distributions for RPKM-normalized DNA ratios; 3) the SDs used for thresholding are significantly higher than any difference in data mean centers. Thus, there is no evidence that misnormalization of the data has dramatically affected our results. The point raised by this reviewer about capturing technical biases in Illumina data is the precise reason why we compare relative mRNA abundance measured in each strain pair to relative DNA abundance measured in the same way (see more below). We again highlight that the significant enrichment for functionally related genes among the selected group strongly suggests that we have not selected random genes, but rather enriched for functionally related groups.

2) Paragraph four, subheading “A substantial fraction of amplified genes show lower-than-expected expression”: Definition of the cut off is the critical point. Why the cut off for mRNA levels should be based on DNA abundance is beyond our understanding – the variance of DNA abundance and variance of RNA abundance are usually very different. The rationale is not explained.

There is simply no other way to do this analysis – it would be wholly inappropriate to compare a measured value for mRNA (subject to both systematic and stochastic technical variation) to an expected value for DNA abundance. In fact, we would have identified many more genes if we had used an expected ratio of DNA abundance that does not capture measurement biases that can compress measured ratios. We added a statement to more clearly outline our rationale: “Comparing measured mRNA ratios to measured DNA ratios (as opposed to theoretical DNA ratios) is critical to capture systematic and stochastic variation in the technical measurements. Our thresholding method allowed us to account for gene-specific biases in sequencing counts while incorporating measurement noise (and minimizing sequencing costs).”

3) The next part of the paragraph brings a flurry of numbers that is difficult to follow. They first state that they identified 163 of 882 genes tested to be dosage compensated – and not 838, as stated by Torres. But in the Hose et al. manuscript, there is indeed the number 838, and the number 163 or 882 is nowhere to be found. They also claim that their subsequent analysis defined stringent criteria for dosage compensation to be met by 88 genes out of 738. They cite Figure 4A in Hose et al., but again, this number is not there. This means dosage compensation in 12% of amplified genes, but at 15% FDR. None of these numbers was mentioned in the previous manuscript.

We have attempted to clarify this section by outlining in paragraph one, subheading “A substantial fraction of amplified genes show lower-than-expected expression” the three sets of expression analyses done in Hose et al. The first compares aneuploid to non-isogenic euploid strains and identified 838 out of 2,204 (38%) genes with lower-than-expected expression in aneuploid strains, based on biological duplicates. However, these 838 genes include those responding to the aneuploidy and genes whose expression is affected by heritable polymorphisms across the non-isogenic strain pairs. We were careful to explain in Hose et al. that this group does not represent dosage compensated genes. Torres et al. imply this and apparently perform their analysis on this gene set.

We then outline the two analyses we did to call dosage compensated genes: the first compared biological duplicate measurements across three aneuploidy-euploid strain pairs (179 of 882 genes (20%) met these criteria, Hose et al. Figure 4B). The second compared biological triplicate measurements across two strain panels using the MLR model (172 out of 773 genes (22%) met the relevant classification, Hose et al. Table 1). We then cite how the genes from the two analyses were combined: because the paired-strain analysis was less stringent (owing to duplicated instead of triplicated replicates), we required that the genes also be identified from the non-isogenic strain comparisons in Figure 4A, in other words the genes had to pass the threshold in four biological replicates. This left 73 of the 179 genes that were added to the 172 genes from the MLR analysis, for a total of 245 genes for downstream analysis (as outlined in Hose et al. main text). We hope this has clarified our analysis without muddying the text with more numbers.

4) Paragraph five, subheading “A substantial fraction of amplified genes show lower-than-expected expression”: Why the SD across all transcripts is not relevant is beyond our understanding. Both SDs across replicates and SD across all transcripts within one sample are relevant, each of them provides different information.

Torres et al. cite the SD of RPKM values across all transcripts in the cells – transcript abundance across the transcriptome varies several orders of magnitude, and thus the SD of all RPKM values is very large (as it should be). The reason this SD is not relevant is that we never used raw RPKM values for any of our work, but rather the relative RPKM in each aneuploid versus the isogenic euploid. As we presented, the SD of the replicate mRNA ratios and the SD across all mRNA ratios on each affected chromosomes are in the same range as the SD of the DNA ratios. The SDs are certainly important – but only the SDs of the data types being studied.

5) Paragraph six, subheading “A substantial fraction of amplified genes show lower-than-expected expression”: The criticism of MLR by Torres and Amon was based on the fact that only 3 samples were used for the analysis, which is not sufficient; here it is claimed that 12 samples were used – however, this number includes the biological replicates of 3 samples only, which is indeed not sufficient. All the additional analysis does not help the fact that the model is based on 3 samples.

The important point is that the MLR model is fit to all of the data for a given panel at once, which includes three ratios for each gene in each of three strain comparisons plotted against the measured DNA. We argue that fitting a linear model across nine relative mRNA measurements and comparable DNA measurements per gene is as accurate a method as we can think of to define the genes we’re interested in.

*6) They agree with Torres and Amon that the distribution of the gene expression of amplified genes is normal (not skewed), yet argue that the downregulated genes represent dosage compensation, whereas the upregulated genes are a part of the general response to aneuploidy* – *it is very unclear why each of these two outliers' groups should be interpreted differently. Their argumentation in the chapter "Unique features of selected genes...." is a prime example of scientific tautology: genes are dosage compensated because it is disadvantageous for the cells to overexpress them.*

Our argument for this was based on data: the group of lower-than-expressed genes we identified is enriched with statistical significance for genes that are toxic when over-expressed in the lab strain (see Hose et al. for references). These genes are also enriched for genes known to be dosage compensated (see paragraph two, subheading “Unique features of selected genes suggest the group is enriched for dosage compensated genes”). The genes with amplified expression are explained by known effects of ploidy on cell size (Wu et al. 2010).

Accordingly, for these genes a propensity to CNV is observed. They do not observe propensity to CNV for genes from these categories when they are present in normal copy numbers – but this is most likely because they are not overexpressed and therefore there is no need to select against them! They also never say what gene categories are downregulated among unamplified genes (likely the same categories as on the amplified chromosomes).

It is not entirely clear what this reviewer is referring to, but perhaps he or she is misunderstanding the analysis. We quantified CNV based on publicly available datasets of *S. cerevisiae* strains, then simply analyzed trends across the different gene groups we defined here and in Hose et al. The genes we identified with lower-than-expected expression show, as a group, a higher propensity for gene duplication in publicly available datasets measuring CNV. With regard to the second point, in this work we removed from consideration all amplified genes belonging to functional groups enriched among the unamplified genes with a significant expression difference in that strain (see subheading “Unique features of selected genes suggest the group is enriched for dosage compensated genes” and Methods). Aside of mitochondrial genes, most of the other functional categories reported in Hose et al. remain enriched in our gene group. The enrichments remain significant when each chromosome is held out of the enrichment analysis, showing that the effect is not driven by a particular strain/chromosome. The enrichments are not significant for the set of unamplified genes that are repressed in multiple strains as part of a common aneuploidy response (Hose et al.).

*7) There are also genes that are up-regulated – the identified categories broadly agree with the previous gene categories identified to be upregulated in response to aneuploidy (they cite the work from Fink lab, which relates to tetraploidy, but see more relevant Amon, Torres, and Storchova lab results). Remarkably, the dosage compensated genes closely match the categories identified previously to be downregulated in response to aneuploidy – ribosome biogenesis, translation, transcription* – *in several different species (see Amon, Torres, Storchova, Foijer laboratories). This option should be considered.*

Respectfully, this reviewer is incorrect. The work by the Fink lab showed that fully tetraploid cells are bigger and have higher expression of genes encoding cell-surface proteins – there is no mention of anything related to ribosome biogenesis or translation factors in that work. To clarify one of our main points in this work: genes encoding ribosomal proteins (RPs) and translation factors can be repressed as a coherent group as part of the environmental stress response – indeed, the strains studied by Amon and Torres are very stressed and show robust activation of the ESR, including down-regulation of the entire group of RP genes and translation factors defined in the ESR, whether those genes are amplified or not. Our results are distinct in that a) the ESR as defined is not activated in our strains and b) the reduced expression is largely specific to the amplified genes. The more likely explanation that we favor is that at least some of these genes, especially RPs that are known to be dosage compensated by feedback, are regulated in response to their own gene copy number.

8) In paragraph five, subheading “Unique features of selected genes suggest the group is enriched for dosage compensated genes”, they cite a paper by Springer et al. (Mol Sys Biol 2010) in support of the presented work. But Springer's manuscript is focusing entirely on dosage compensation on protein level, whereas Hose considers mRNA; of individual proteins only, measured one-by-one by FACS, whereas Hose et al. considers whole chromosomes only, and Springer analyzes loss of copies, whereas Hose analysis gain of gene copies.

We have added a clarification to the Discussion that Springer used a GFP reporter with a non-native 3’UTR for that work.

In conclusion, the arguments do not seem strong enough to justify their analysis approach. Secondly, even if we would agree with the analysis, they found that the mRNA of 12 – 13% of genes might be dosage compensated at a FDR of almost 15% – a result that in fact says that there is no general dosage compensation.

We respectfully submit that the numbers cited in this work are on the same order as the numbers claimed in the original Hose et al. manuscript. As outlined in the last paragraph of this manuscript, we did not intend to claim that dosage compensation was widespread or functioned at most genes, and we never used the term. One of our key points in the original work and in this paper is that dosage compensation likely plays an important role in evolution, particularly in facilitating CNV. All of our work remains consistent with this notion.

Reviewer #2:

1) As I already indicated in my review of Amon et al., I am also puzzled by the non-integer changes in sequencing depth. I do think that this suggests that the strains could indeed be unstable. However, I also noted that we need to be sure that sequencing biases are not involved. I suggested a few relatively easy experiments to verify whether the populations are homogeneous and whether they show the expected aneuploidies. It seems easy enough to measure the copy number of a few key regions in a few key strains using qPCR, and to do this on a few different (small) colonies of the same culture. This would close the discussion.

We thank the reviewer for this suggestion. For five of the strains we worked with, we have done a similar analysis, first at the culture level and finally on individual colonies from a passaged culture, that shows that in most cases the aneuploidies are quite stable (but can be lost stochastically, see text for details). Therefore, we have opted not to add more of these comparisons at this time.

2) I follow the arguments about the error calculations. Although I am not an expert in this area, I think that, Gasch, Hose and coworkers indeed used correct calculations, but their sample size (number of replicates) are very low. Again, a few extra experiments using qPCR would help to close the discussion.

We provided qPCR and DNA microarray analysis of several genes from Chr12 in Hose et al. It is true that to measure very small, but real, differences in expression would require more than biological triplicates of mRNA data.

*3) The number of genes that may show dosage compensation (245) seems quite low, especially since many of these might be false (false discover rate*…*). Even if some GO enrichment is found in this set, it is not clear to me that the number is high enough to call this "widespread". My best guess is that strains become aneuploid and the relatively quickly acquire mutations that normalize the expression of a few key genes for which changes in expression are not tolerated. Whether this is special for wild yeasts and should be called "dosage compensation" is a difficult question (obviously, the dosage is compensated, but using the name also suggests a general, dedicated mechanism, and this may very well be lacking*…)

Respectfully, we never used the term widespread (that was used extensively by Torres et al.). Indeed, we do not believe that dosage compensation functions at most yeast genes. Our main interest in this manuscript was to show that genes potentially subject to dosage compensation display unique evolutionary patterns. We agree that there is unlikely a single or universal mode of dosage compensation analogous to the silencing of sex chromosomes, for example, and we did not indent to claim that in our original manuscript. With regard to the acquisition of rapid mutations that down-regulate genes: this was the main motivation for deriving the isogenic strain panels, since heritable polymorphisms that down-regulate genes as part of an adaptive response would be detected in our analysis. It is certainly possible that this occurs at some of the genes with heritably reduced expression that we identified in Hose et al., but it cannot explain the reduced expression per gene copy in our isogenic strain groups.

One notable (anonymized and lightly edited) comment from the reviewers’ discussion:

On the second issue, the extent of dosage compensation, I think that there is a fundamental disconnect between the groups in how to assess dosage compensation. Gasch (in Hose and Gasch) seek to take a gene specific view that accounts for gene-to-gene variability. This is noble and philosophically sound, but fraught with problems they don't address – namely that they have way too small a sample size to *really* do this (as also noted by both other reviewers). The Amon approach is to take a more distribution-based approach – which is, in principle, better in low sample sizes – but as Gasch correctly points out (and was a major criticism of the Torres manuscript) – the Torres methods are so poorly described as to be impossible to assess. Even if you think the Gasch method has merit, in the end they do back peddle on the number of dosage compensated genes (in Gasch compared to Hose) leading one to suspect we are talking about something that is either a small class or even non-existent.

It is true that our group versus Torres et al. have a philosophical difference in describing dosage compensation: we have focused on gene-level analysis without implying mechanism; they have focused on chromosome-wide effects. In this regard, both our groups agree that there is no evidence for a chromosome-wide mechanism of dosage compensation. We do believe that we have statistical power to identify genes subject to dosage compensation, and the fact that the gene groups we identified are enriched for myriad features of interest reveals that our results simply cannot be explained away as noise. Our more stringent analysis in this work presents a restricted gene set, which while on the lower end of the range we cited in Hose et al. remains consistent with our original claims. Fundamentally important to us are the evolutionary signatures and functional enrichments among the gene set we identified – while the numbers may appear small depending on perspective, it is the impact on evolution that we center on.